# Gene Therapy of Adrenomyeloneuropathy: Challenges, Target Cells, and Prospectives

**DOI:** 10.3390/biomedicines13081892

**Published:** 2025-08-04

**Authors:** Pierre Bougnères, Catherine Le Stunff, Romina Aron Badin

**Affiliations:** 1Laboratoire des Maladies Neurodégénératives, MIRCen Institute, Commissariat à l’Energie Atomique, 92260 Fontenay-aux-Roses, France; catherine.le-stunff@inserm.fr (C.L.S.); romina.aron-badin@cea.fr (R.A.B.); 2Therapy Design Consulting, 94300 Vincennes, France; 3UMR1195, Inserm and University Paris Saclay, 94270 Le Kremlin-Bicêtre, France

**Keywords:** gene therapy, AAV, adrenomyeloneuropathy, spinal cord, axonopathy

## Abstract

Gene replacement using adeno-associated viral (AAV) vectors has become a major therapeutic avenue for neurodegenerative diseases (NDD). In single-gene diseases with loss-of-function mutations, the objective of gene therapy is to express therapeutic transgenes abundantly in cell populations that are implicated in the pathological phenotype. X-ALD is one of these orphan diseases. It is caused by *ABCD1* gene mutations and its main clinical form is adreno-myelo-neuropathy (AMN), a disabling spinal cord axonopathy starting in middle-aged adults. Unfortunately, the main cell types involved are yet poorly identified, complicating the choice of cells to be targeted by AAV vectors. Pioneering gene therapy studies were performed in the *Abcd1^-/y^* mouse model of AMN with AAV9 capsids carrying the *ABCD1* gene. These studies tested ubiquitous or cell-specific promoters, various routes of vector injection, and different ages at intervention to either prevent or reverse the disease. The expression of one of these vectors was studied in the spinal cord of a healthy primate. In summary, gene therapy has made promising progress in the *Abcd1^-/y^* mouse model, inaugurating gene replacement strategies in AMN patients. Because X-ALD is screened neonatally in a growing number of countries, gene therapy might be applied in the future to patients before they become overtly symptomatic.

## 1. Introduction

X-ALD was in the past considered very rare (1/50,000–1/100,000) [1,2,3,4], but its true incidence assessed by neonatal diagnosis in the US population averages 1/10,000 [5]. It is the only NDD screened at birth since 2013 in a growing number of countries, using the detection of elevated VLCFA concentration in plasma, the disease hallmark [5,6,7,8,9,10,11,12,13,14,15,16]. Neonatal diagnosis is already performed in 43 American states [5,6,16,17] and in the Netherlands, Italy, and Japan [12,18]. It will probably become widespread in many other developed countries. In the future, it may thus become possible to apply gene therapy at the earliest stages of AMN, possibly before clinical manifestations. Until then, gene therapy should be applied with priority to adult patients who present overt clinical manifestations.

To rescue monogenic loss-of-function diseases, the strategy of gene therapy is to replace the mutated gene with a normal gene in the cells that are crucial to disease mechanisms. The target is sometimes obvious, for example, the motor neurons in the anterior horn of the spinal cord in infants affected by SMA [19]. In contrast, defining the cellular targets of gene replacement in AMN remains a major challenge since neurons, axons, astrocytes, OL, and microglia can all be suspected of participating in the pathophysiology [20].

This review will dissect the information that gene therapy experiments using AAV vectors in *Abcd1^-/y^* mice have provided as a prerequisite for developing an effective treatment of AMN.

## 2. Clinical Manifestations of AMN

The two different forms of X-ALD both affect the white matter (“leukodystrophy”) of the central nervous system (CNS). The most dramatic is the cerebral demyelinating form (cALD), which affects male children and leads to their death within a few years. The other, more frequent, adult form of X-ALD, identified at the end of the 1970s [21,22,23], is adreno-myelo-neuropathy (AMN). In its “pure” form, AMN affects only the long tracts of the spinal cord, giving a progressively disabling spastic paraparesis of the lower limbs that begins in middle-aged male adults and progresses inexorably over the next decades [24,25,26,27] and is associated with ataxia, pain in the legs, impotence, and voiding abnormalities, whereas deficits in arms and hands remain mild [28,29]. We now know that patients who survive cALD thanks to hematopoietic stem cell transplantation (HSCT) develop AMN in adulthood [30].

AMN does not remain pure in all patients. Indeed, progressive brain demyelination (cerebral AMN, cAMN) occurs in one-fifth of them [31], resulting in cognitive deficits, psychomotor manifestations, and loss of visual memory [24,32]. A confluent brain demyelination [33] with severe deterioration of brain functions even occurs in 5% of AMN patients [34,35]. Although an X-linked disease, AMN gives slowly progressive symptoms of myelopathy in two-thirds of heterozygous women [33,36,37,38,39].

A number of AMN patients have electrophysiological manifestations of peripheral neuropathy [40], the symptoms of which are masked by the prominent myelopathy syndrome. In addition, 3/4 of patients have hormonal evidence of adrenal insufficiency at the time of diagnosis [41,42,43].

## 3. The Neuropathology of AMN

The design of an efficient gene therapy approach for AMN requires a deep knowledge of its neuropathology. In its purest and most frequent form, AMN is primarily and almost exclusively a disease of the spinal cord white matter, a “leukomyelopathy”. To our knowledge, early lesions of myelopathy have not been reported. The only available neuropathology data came from ten adult male patients who were autopsied in the 1970s at terminal stages of AMN [21,22,24,34,44,45,46,47], ten to twenty years after diagnosis. All showed spinal cord atrophy associated with the degeneration of long ascending and descending tracts.

Axon degeneration and myelin loss are most severe in the cervical fasciculus gracilis, and dorsal spinocerebellar and lumbar lateral corticospinal tracts [24]. The ascending fasciculus gracilis originates from first-order neurons located in dorsal root ganglia (DRG). Its heavily myelinated axons convey conscious proprioceptive, tactile, and vibratory information from the lower trunk and extremities to the sensory cortex through intermediary synapses in the gracilis nucleus and the thalamus. In AMN patients, the fasciculus gracilis shows a prominent loss of myelinated fibers and OL [45], associated with axonal atrophy and very thin or disintegrating myelin sheaths [24,46]. These lesions predominate at the cervical level. Two other tracts are also affected: the ascending dorsal spinocerebellar tract at the cervical level and the descending lateral corticospinal (pyramidal) tract at the lumbar level. Remarkably, this axonal degeneration reaches its maximum far from the neurons of origin that are located either in the DRG (ascending tracts) or the cerebral motor cortex (descending tracts). Because the cell bodies of these parental neurons are intact and in normal number [24,48], AMN is considered a primary “dying-back” axonopathy, with the longest spinal cord axons suffering the greatest damage. Microglia are present in lesions, but blood-borne inflammatory cells are absent, so that AMN myelopathy is considered a non-inflammatory process. Gray matter is unremarkable [24].

Gene therapy vectors that aim at targeting neurons would thus need to reach two neuronal populations at the cell body level. The first-order neurons of the fasciculus gracilis or spinocerebellar tracts should be transduced by bringing vectors into the DRG that are located below the T6 spinal cord level, which is possible with intrathecal injections [49,50,51,52,53,54]. Concomitantly, the vector should also target the first-order neurons of the corticospinal tract located in the cerebral motor cortex, a region of the adult CNS that has been poorly transduced by the common routes of administration of AAV vectors [55]. To correct both populations of neurons, gene therapy would thus require two separate routes of administration.

In addition to the three previously cited structures consistently showing atrophy at a terminal stage of AMN, less severe axonal abnormalities are observed in other tracts of the spinal cord, brainstem, or cerebellum.

Apart from the CNS, studies of peripheral nerves in AMN [21,22,45,46,56,57,58] have shown variable and mild lesions of chronic axonal atrophy without inflammation [24,59]. Inclusions in Schwann cells were observed in the peroneal nerves [59].

Even if ~1 out of 5 AMN patients develop an inflammatory demyelination of the brain [24,34,35], the gene therapy of cerebral AMN is not included in the current review.

## 4. The *ABCD1* Gene

*ABCD1*, the gene responsible for X-ALD (OMIM300100), was identified in 1993 [11]. A recent search in the *ABCD1* registry (https://adrenoleukodystrophy.info, accessed on 1 July 2025) [20] found 172 variants annotated as “probably pathogenic” and 825 as “pathogenic”. The 2.3 kb long coding sequence can be packaged in the capsid of an AAV vector.

The ALD protein (ALDP) is an ATP-binding cassette transporter located in the peroxisomal membrane [60]. ALDP transports VLCFA into peroxisomes to be degraded through beta-oxidation [61,62,63,64,65]. In humans, two other ABC transporters shuttle VLCFAcyl-CoA into peroxisomes: adrenoleukodystrophy-related protein (ALDR) coded by *ABCD2* and PMP70 coded by *ABCD3* [64,66,67].

To date, ALDP expression has not been studied or reported in normal human spinal cord. In the adult brain, ALDP expression is found in astrocytes and microglia in subcortical and cerebellar white matter. Expression is high in a subset of OL in the corpus callosum, internal capsules, and anterior commissure [68] and weak in other OL populations [69]. It is sparse in neurons of the cerebral cortex, including the motor cortex and sensory cortex, and absent in neurons of the cerebellum, putamen, caudate, and substantia nigra [69]. Forty percent of neurons in a thoracic DRG show ALDP expression [69].

## 5. Cell Types Involved in AMN Pathology

The understanding of AMN pathogenesis continues to face numerous unknowns, notably the identification of the cell types that contribute to the white matter pathology in the spinal cord. This section will thus only briefly review the different cell types that could be potential targets for gene therapy. Not only should target cells play a critical role in neuropathology, but they should also maintain vector genome expression in the long term. Indeed, to persist durably in transduced cells, episomal transgenic *ABCD1* copies should not undergo mitotic dilution. Poorly dividing or post-mitotic cells are thus the distinctive targets of AAV gene therapy [70].

### 5.1. Oligodendrocytes

For a number of reasons recently reviewed in [20], we believe that spinal cord interfascicular myelinating OL are crucial target cells for AMN gene therapy. Their close connection with axons is the main reason. OLs are abundant alongside long axon tracts, each of them enwrapping tens of axons with myelin sheaths [71,72]. To make myelin, OLs synthesize lipids including VLCFA [73,74] and degrade large quantities of VLCFA in their peroxisomes [73]. The OL content of peroxisomes [75,76,77] is several times higher than that of neurons or astrocytes [73,75,76,78,79], making OL highly vulnerable to the peroxisomal lack of ALDP. In addition, studies in Drosophila knockouts revealed that, when VLCFA accumulate in myelinating glial cells (the orthologs of OL) [80], they can be converted into sphingosine-1-phosphate (S1P) [81], a toxic lipid able to poison fly axons [80] (this mechanism of axonal degeneration has not been tested yet in a mouse model). Another prominent role of OL is to support axons by funneling energetic substrates, such as glucose, glycolytic substrates [82,83], and ketone bodies [84,85] directly to the periaxonal space through myelinic channels [72,86,87]. Last but not least, mature OLs have a long lifespan and limited renewal rate in humans and mice [88,89,90,91], which should *a priori* allow vector copies to persist durably in their nucleus.

### 5.2. Neurons

Neurons have remained the main target cells of gene therapy for the majority of NDD and as such, for AMN. Neurons face both an enormous energetic demand [92] and a problem of energy distribution, given that the average axon is ten thousand-fold longer than its parental cell body. The myelinated axons of the long spinocerebellar and corticospinal tracts are thus the most energetically vulnerable. Notably, the neurons of the motor cortex convey energy-demanding motor messages to lumbar synapses, through the pyramidal tract, over 65 cm in humans (less than 5 cm in mice). Axons rely on astrocytes for lactate supply and on OL for glucose, lactate, and fatty acid supply. Mitochondria are recruited to the end of long axons to allow oxidative energy provision for axonal transport. Thus, AMN axons gradually degenerate toward the cell body [93,94] (“dying-back axonopathy”), likely due to an energy crisis [95,96]. Peroxisomes are few in neuron perikarya or dendrites, rare within axons [75,76,77,78,97,98], and absent in synaptic terminals of the adult spinal cord [77]. Because normal neurons and axons do not degrade VLCFA significantly [99], they are not exposed to intrinsic VLCFA accumulation. This is why restoring peroxisomal transport of VLCFA through *ABCD1* gene transfer in the cell body of parental neurons is not likely to rescue axonal degeneration.

### 5.3. Microglia and Astrocytes

Although not primarily involved in axonal degeneration, spinal cord microglia and astrocytes may protect degenerating axons from triggering deleterious inflammatory processes. Microglia live 6 to 41 months in the mouse brain [100] and are renewed several times during a human lifespan [101,102]. Peroxisomes are abundant in microglia [73,75,103] and brain microglia express abundant ALDP [68]. Microglia are continuously monitoring what happens in the CNS [104], and support myelin formation and integrity over their whole life [105,106,107]. The fact that microglia function declines with age [108] is also important for AMN and NDD manifesting in late adulthood.

Astrocytes are highly plastic cells that differ in various regions of the CNS (reviewed in [109,110]). As a double-edged sword [111], reactive astrocytes can either be homeostatic, adaptive, and beneficial, or maladaptive and detrimental [112], losing their neuro-supportive role [113] or driving pathological mechanisms through the release of pro-inflammatory cytokines [114,115]. Dying neurons release ATP and potassium, which can activate astrocytes [116]. Astrocytes elongate fatty acids into VLCFA with Elovl1 [117]. Astrocyte peroxisomes carrying VLCFA degradation are localized in cellular processes including the end-feet [73,76,118]. In the healthy adult CNS, astrocytes are post-mitotic cells that rarely divide [119], allowing the persistence of AAV vector copies in their nucleus.

Whether *ABCD1* mutations can change the functions of microglia and astrocyte substates and their relationship with degenerating axons may be key to understanding disease mechanisms. Furthermore, if subpopulations of spinal cord microglia or astrocytes of AMN patients were lacking physiological neuroprotective functions, *ABCD1*-based gene therapy targeting these cell types would be beneficial.

### 5.4. Other CNS Cells

Macrophages and blood-borne immune cells do not appear to play a significant pathogenic role in the absence of local inflammation in the spinal cord of AMN patients.

Endothelial cells, as BBB constituents, abundantly express ALDP [68,69,120] and are extremely important in cALD pathology [121]. In contrast, AMN patients have no BBB breach and no invasion of circulating monocytes or lymphocytes into their spinal cord. Endothelial cells thus do not appear as major targets of gene therapy.

### 5.5. Schwann Cells

Schwann cells synthesize fatty acids including VLCFA to build myelin sheaths for enwrapping peripheral axons. Since sural or peroneal nerves may show inclusions in Schwann cells and abnormalities of large myelinated fibers [59], it is likely that Schwann cells play a central role in the degenerative neuropathy affecting AMN patients and could be specific targets for gene therapy.

### 5.6. Cells of the Adrenal Cortex

Primary adrenal insufficiency is common in AMN and other peroxisomal disorders [122,123]. Chronic VLCFA and cholesterol accumulation are believed to trigger apoptosis of adrenocortical cells, leading to a decrease in cortisol production [124].

## 6. *Abcd1* Knockout Mouse Model of AMN

An *Abcd1* knockout (KO) created in the late 1990s mimics the VLCFA accumulation in the spinal cord and the late and progressive spinal cord axonopathy of pure AMN [125,126,127]. Because the *Abcd1* gene is located on the X chromosome, all studies have used only *Abcd1* KO males (*Abcd1^-/y^*) for homogeneity. Our follow-up study of dozens of *Abcd1^-/y^* mice until 2 years of age [128,129] confirmed and extended previous findings [126,127,130,131]. Because all trials of AMN gene therapy using AAV vectors have been carried out in *Abcd1^-/y^* mice, it is worth recalling their main characteristics at a phenotypic, cellular, and biochemical level.

### 6.1. Motor Phenotype

While there are no established rules for the demonstration of gene therapy efficacy, we believe that the best criteria are clinical and based on a significant improvement of motor deficits. Left untreated, *Abcd1^-/y^* mice develop progressive, late-onset manifestations that closely resemble the paraparesis of AMN patients, with motor and balance defects predominating in the hind limbs and tail. Motor tests include the rotarod, the crenellated or non-crenellated bar/beam walking test, the edge test, the hind limb extension reflex/hindlimb clasping test, the spontaneous open field movement test, the gait pattern test, the footprint pattern test in tunnel walking, and the spontaneous rearing test. Because only a minority of mice show a truly severe neurological deficit [128], the degree of deficit is difficult to quantify in the others. Observations in our studies [128,129] and in the literature [132] show that behavioral outcomes are highly variable both individually and between mice in the same group, declining with age, and quite difficult to quantify between groups of mice. Indeed, the scores assigned to the tests reflect the clinical deficits rather grossly, and do not dissect out the ataxic and motor components. The tests used are not consistent across publications and show variable degrees of sensitivity and reproducibility. To compare groups of mice, we find it more reliable to monitor the individual decline in motor performance at 16 and at 24–28 months. To gain confidence, the tests should be repeated individually at each age, as training is important. We also recommend that three experimenters, blind to the genotype of the mice, analyze separately the videos taken during the tests [128]. The distribution of individual responses to tests is Gaussian [128], as is the distribution of the ages at which the first deficits appear [20,128], making it risky to compare small numbers of mice, as in some publications [49,132,133,134,135]. It is also desirable to compare *Abcd1^-/y^* and wild-type (wt) littermates reared in the same cage to reduce environmental heterogeneity. Last but not least, a normal decline in motor performance with age, well visible in wt mice, complicates the analysis.

In our study of 40 *Abcd1^-/y^* mice, the response to motor tests remained normal until 16 months of age. Motor and coordination deficits predominantly affecting the hind limbs and tail appeared in ~25% of *Abcd1^-/y^* mice at 18 months [136]. Motor deficits strike more mice and worsen between 18 and 24 months of age. At 24 months, ~70% of mice had obvious motor manifestations, but 33% failed to show clinical pathology [128]. This late and variable evolution of motor deficits creates difficulties for the evaluation of gene therapy effects, which thus requires long-term studies and a large number of mice [128].

### 6.2. Neuropathology in Abcd1^-/y^ Mouse

Because *Abcd1^-/y^* mice have not been followed until terminal stages and death, it is not known whether corticospinal and spinocerebellar tract atrophy would be observed, as in the postmortem spinal cord of AMN patients. OL and astrocyte counts are normal in the dorsal columns of the spinal cord, corpus callosum, and cerebellum [128,129]. Electron microscopy (EM) found that myelin and axons appear normal in 3- to 4-month-old *Abcd1* KO mice [127]. In 16- to 21-month-old *Abcd1^-/y^* mice (*n* = 4), the cervical, thoracic, and lumbar regions of the spinal cord showed some focal abnormalities in white matter [125], such as redundant myelin sheaths, and degenerating axons [125]. EM revealed «pockets of severe ultrastructural aberrations of axons and myelin» [125], and mitochondrial abnormalities were reported in axons [137,138]. In the dorsal columns of the spinal cord of two severely symptomatic *Abcd1^-/y^* mice aged 24 months, we observed β-amyloid precursor protein (APP) clusters suggestive of defective axonal transport, coexisting with normal morphology and thickness of myelin sheaths (unpublished). However, a quantitative and qualitative characterization of axonal pathology in *Abcd1^-/y^* mice is still lacking and would greatly benefit from recent techniques of large-field-of-view imaging at a single-axon resolution, and automatic software to segment datasets [139,140]. This would be particularly important to evaluate the effects of gene therapy approaches on axonal degeneration.

The *Abcd1^-/y^* mouse shows no cerebral pathology [125,126,127] or electrically detectable peripheral neuropathy [141]. No brain or spinal cord lesions are visible macroscopically or by light microscopy, and no significant demyelination or neuroinflammation is detected by immunohistochemistry.

### 6.3. VLCFA Accumulation in the CNS of Abcd1^-/y^ Mice

It is the major biochemical hallmark of the *Abcd1* knockout mouse model. Indeed, the brain of 3- to 4-month-old *Abcd1^-/y^* mice displays elevated C25:0, C26:0, C26:1, C28:0, and C28:1 fatty acid content. Notably, C26:0 levels are 3.7–8-fold normal [74,126,127]. C26:0-carnitine levels are increased ten-fold in the brain and 9-fold in the spinal cord [74]; the C26:0/C22 ratio was 3.8–4.6-fold normal [49,126]. The C26:0-lysoPC content of the spinal cord and brain of 24-month-old *Abcd1^-/y^* mice is 8.4-fold normal and 24-fold normal, respectively [128,129]. The decrease in VLCFA accumulation will be a hallmark of the restoration of peroxisomal degradation by *ABCD1* gene therapy. It remains intriguing that despite this massive accumulation of VLCFA, the brain shows no pathology.

### 6.4. Limitations of the Abcd1^-/y^ Mouse Model

Humans and mice differ in their genetic background and modifier genes, levels of gene expression, specific enzymatic equipment for VLCFA metabolism, biology of cell subpopulations, and neuronal activity in specific CNS regions. Because the *Abcd1^-/y^* mice have a short lifespan, spinal cord axons are exposed to deleterious mechanisms for a much shorter duration than in AMN patients. Aging of neural functions and potential environmental encounters could also play a distinctive role in AMN patients, not in mice. The shorter mouse axons may be less vulnerable than those of patients. Finally, it will be important to follow motor performances and study spinal cord white matter with classical microscopy and electron microscopy (EM) beyond two years of age.

CRISPR/Cas9-based genome editing was recently used to generate a faithful mouse model of a rare microgliopathy bearing human mutations, while knockout of the entire gene had failed to adequately recapitulate the human phenotype [142]. By analogy with this leukoencephalopathy, it would be interesting to use the same technology to generate an X-ALD mouse model harboring human hotspot mutations of *ABCD1*.

## 7. How to Evaluate Gene Therapy Effects in *Abcd1^-/y^* Mice

While other *Abcd1* knockout non-mammalian animals exist [143,144], the *Abcd1^-/y^* mouse is indisputably the most reliable model of AMN for validating and documenting the efficacy of gene therapy approaches.

Before reviewing gene therapy studies that have been performed in the *Abcd1^-/y^* mouse model, it is worth mentioning the methodological principles, conditions, and criteria that should be met for analyzing gene therapy approaches in this model.

### 7.1. Analyzing Gene Therapy Attempts to Rescue the Pathology of Abcd1^-/y^ Mice

#### Sex

Gene therapy studies were performed in *Abcd1^-/y^* male mice for a number of reasons [49,128,129,132,134,145]. The main one was to mimic human AMN. Also, sex dimorphism of the brain [146,147,148] is observed for neurons [149], astrocytes [150], microglia [151], and OL [152,153]. Notably, the density of OL in the mouse spinal cord is 20–40% greater in males. There are also large sex differences in OL and myelin turnover [154]. Androgens upregulate myelin-related genes [155]. Not only can sex steroids make a difference, but unidentified modifiers might exist on the X or the Y chromosome that could influence *Abcd1^-/y^* phenotypes or the response to gene therapy due for example to differences in AAV tropism between males and females [156].

### 7.2. Prevention or Improvement of Motor Deficits

The main criteria for judgment of gene therapy efficacy in *Abcd1^-/y^* mice should include a significant and consistent improvement on several motor tests. The main caveats for interpreting these tests have been cited in Section 6.1. Motor tests include the rotarod, the crenellated or non-crenellated bar/beam walking test, the edge test, the hind limb extension reflex (hindlimb clasping) test, the spontaneous open field movement test, the gait pattern test, the footprint pattern test in tunnel walking, and the rearing test. By watching hundreds of crenellated bar test videos over the years, we found that the motor deficits of *Abcd1^-/y^* mice predominate in the rear train, hind legs, and tail, mimicking the lower body deficits observed in AMN patients. Notably, many *Abcd1^-/y^* mice have perturbed tail balance during this test, while normal mice maintain stability by either lowering their tail position and keeping it static and straight so as to lower the whole-body mass or by extending the tail to the side. Recent studies provide sensitive tests and accurate metrics for specifically studying balance in rodents [157,158] and should be incorporated into the testing of *Abcd1^-/y^* mice. Among the widely used tests, we found the latency to fall to be a robust quantitative metric of *Abcd1^-/y^* axonopathy, even if the rotarod test involves a strong participation of the front legs versus the hind legs.

### 7.3. Cell Transduction

Cell transduction by the vector, sometimes described by adjectives such as widespread, abundant, sparse, etc., should be evaluated by precise measurements. The most commonly used of these measurements is the percentage of transduced cells among a given cell-type population characterized by its specific immunofluorescence biomarkers. The tagging of the *ABCD1* transgene with HA allows a direct measurement of the total cell transduction (HA+/DAPI+), the percentage of a given cell type among total cells, and the percentage of transduced cells among a given cell-type population, three important criteria to determine vector distribution. Needless to say, the studied region should always be specified, whether it is an entire spinal cord piece, or a spinal cord white matter piece, or dorsal funiculi or other spinal cord tracts, and the level of the spinal cord precisely indicated (Figure 1). Expression of *ABCD1* should be quantified in the long term to ensure the durability of transgene expression.

The number of vector genome copies per diploid genome (VGC) can also be used to measure *hABCD1* copy content and persistence in the spinal cord and other CNS regions [145]. RT-PCR can measure *ABCD1* mRNA in nervous tissue [145]. However, both VGC and mRNA copies are average counts reflecting the highly heterogeneous cellular composition of CNS samples. Since a given transduced cell may contain 1 to 50 vector genome copies, one cannot easily interpret average values of VGC or mRNA in a global specimen of spinal cord, including white and/or gray matter. However, if, for example, the dorsal columns are dissected out, the counted VGC and mRNA will mostly reflect the transduction of interfascicular OL and astrocytes.

Whenever possible, EM should be used to detect changes in myelin sheaths, number, axon morphology, thickness, and clusters of β-amyloid precursor protein that may occur in response to vector treatment. This has not been conducted, however, in any of the published gene therapy studies [49,128,129,134]. Recent techniques of large field-of-view imaging at a single-axon resolution, and automatic software to segment datasets [139,140] would provide an accurate quantitative and qualitative characterization of axonal pathology to evaluate the effects of gene therapies on axonal degeneration.

### 7.4. VLCFA Accumulation in the CNS

The most sensitive parameters of VLCFA accumulation are currently C26:0-LPC and C26:0-carnitine [74,159]. Restoration of peroxisomal degradation leading to decreased VLCFA accumulation in the spinal cord is an expected outcome of gene therapy approaches. Measuring whole spinal cord specimens may not be as sensitive as measuring these parameters in white matter, or even better in the dorsal columns of the spinal cord that supposedly contain the most affected axons.

## 8. Gene Therapy Attempts in *Abcd1^-/y^* Mice

Except for early attempts using HSC transplantation, all gene therapy approaches have used neurotropic AAV9 vectors to import and express *hABCD1* copies into targeted cells of the CNS. To prevent or rescue the myelopathy of *Abcd1^-/y^* mice, gene therapy should restore a sufficient level of *hABCD1* expression and activity in cells crucially involved in the function of the affected spinal cord tracts. Pioneer attempts used “ubiquitous” promoters to restore *hABCD1* expression in a variety of neural cells of the spinal cord white matter [49,132,134]. We will see that expression was not ubiquitous, though, in CNS cell types. A more recent approach used another promoter instead to drive *hABCD1* expression in OL [128,129] according to the hypothesis that these cells play a central role in AMN pathogenesis [20].

### 8.1. Early Cell and Gene Therapy Targeting HSC or Microglia

The impressive effects of the first heterologous hematopoietic stem cell (HSC) transplantation in a single cALD patient in 1990 [160] pointed towards a role for brain microglia, because at that time microglia were thought to originate from bone marrow progenitors [161]. In the early 2000s, microglia thus became the first target of gene therapy experiments in *Abcd1^-/y^* mice. Mice received bone marrow CD34+ progenitors extracted from a human donor [162] or from cALD patients that had been transduced with *ABCD1* using an HIV-derived or a lentiviral vector [163,164]. Results showed a rescue of the motor deficits of *Abcd1^-/y^* mice [163] that at the time, the authors enthusiastically thought their approach “would be a valuable alternative both for children and adults with cALD and/or AMN” [163]. The likely reason for these early results published in an abstract to remain unconfirmed is that *Abcd1^-/y^* mice have an intact blood-spinal cord barrier [165] that could allow HSC to enter the spinal cord and improve motor phenotype. The recent finding that surviving cALD patients who have received allogeneic HSC transplantation in childhood subsequently develop AMN confirms that the pathogenesis of AMN spinal cord axonopathy cannot be reversed by ex vivo gene therapy with HSC [166].

However, one should not exclude specific spinal cord microglia subpopulations or substates [167,168,169] from the list of potential targets of interest for the gene therapy of AMN using AAV vectors [170]. Indeed, because some spinal cord microglia subpopulations have shown neuroprotective effects in spinal cord injury [171], one could hypothesize that microglia might contribute to rescuing dysfunctional or degenerating spinal cord axons. This remains speculation, however, because specific microglia transduction has not been explored in this model. Another question is the turnover rate of microglia. While they are long-lived in the mouse brain [100], microglia are renewed several times during a human lifespan [101,102], which would result in a significant mitotic dilution of vector copies [70].

### 8.2. Neuron-Targeting Gene Therapy

The vector used in these studies comprised *hABCD1* cDNA controlled by the so-called ubiquitous promoter CBA, packaged in an AAV9 capsid [49,132,134,145]. We will briefly review these approaches.

#### 8.2.1. Preventive Approaches

Interventions targeting neurons took place before the age at which motor deficits start to appear in *Abcd1^-/y^* mice [49,132,133,134,135]. In these yet asymptomatic mice, gene therapy was thus aimed at preventing, not at reversing, installed neurological deficits. To determine the success of this attempt, it would have been necessary to pursue the observation of vector-treated mice until 24 months of age, when a majority of *Abcd1^-/y^* mice show clear signs of axonopathy.

In a first study, the AAV9-CBA-*hABCD1* vector was administered IV at 1.5–2 months of age (1–3 E12 vg per mouse, *n* = 6) [134] and mice were euthanized only two weeks later. *hABCD1* expression was said to be widespread in neurons of the spinal cord and brain. It would have been of interest to know if neurons of the motor cortex or DRG expressed the transgene (see Section 3). Around 23% of astrocytes, 18% of microglia, 7% of OL, and 65% of vessels in the spinal cord were transduced. The authors did not report whether transduced glial cells were counted in the white matter and/or dorsal columns, and at which cervical, thoracic, or lumbar level. C26:0-LPC was reduced by 40% in the brain and spinal cord. High levels of *ABCD1* expression were found in the liver and adrenals.

These results were compared with the administration of the AAV9-CBA-hABCD1 vector in the left lateral ventricle (1.E11 per mouse, *n* = 12) at 1.5–2 months of age and mice were again euthanized 2 weeks later [134]. As would be expected given the injection procedure, most transduced cells were in the periaqueductal gray matter close to the injected ventricle or in deep basal nuclei [134]. *hABCD1* was poorly expressed in neurons, while 12% of astrocytes and 3% of microglia were transduced in the spinal cord. OL and endothelial cells did not express the transgene. The authors do not report in which region of the spinal cord or white or gray matter transduced cells were counted. In the brain, neurons, microglia, and astrocytes were the main targeted cells versus OL or endothelial cells. Expectedly, ICV injection resulted in a major transduction of the structures located near the injection site, and some transduction was also found in the cerebral cortex, thalamus, and cerebellum. C26:0-LPC was not significantly reduced in the brain and spinal cord.

An alternative approach delivered the AAV9-CBA vector intrathecally (IT) at different doses: 2.E9–2.E10 [145], 2–3 E11 [49], 5.5.E10–5.5.E11 vg/mouse [145], and at ages ranging from 3 to 12 months [132,145]. Transduced cell types were studied at 3.5–5.5 months of age in the early-injected animals [49]. In the spinal cord, *hABCD1* expression was said to be widespread but sparse [49] involving neurons and astrocytes. In contrast, many DRG neurons expressed *hABCD1*-HA [49]. Liver, heart, and adrenals were transduced, notably after an IT bolus [49]. The C26/C22 ratio, or C26 content, decreased by 17% in the spinal cord, but not in DRG [49], and surprisingly increased by 58% in the liver [49].

Note that these studies reported transduction observations soon after vector delivery and thus could not evaluate the persistence of *hABCD1* copies in the various cell types.

#### 8.2.2. Clinical Phenotypes

In other experiments using IV (1–3.E12 vg per mouse, *n* = 4), ICV (1.E11, *n* = 10), or IT (1.E11, *n* = 7) routes of vector administration at 5–13 months, motor tests were performed between 18 and 20 months [132]. Globally, hindlimb clasping revealed no difference between vector-treated and untreated mice, and more surprisingly between untreated *Abcd1^-/y^* mice and wild-type mice [132]. Only ICV-injected mice (*n* = 6) showed better hindlimb clasping at 18 months of age compared with untreated (5–2) or wild-type mice (*n* = 6) [132], but rotarod and bar tests found no benefit [132].

#### 8.2.3. Late Studies in a Separate Group of Vector-Treated *Abcd1^-/y^* Mice

In another study of the [AAV9-CBA-ABCD1] vector, *Abcd1^-/y^* mice were injected IT at 20–23 months of age and euthanized 2 months later [145]. Motor tests were not reported, likely because of the very small doses of vector (2E9–2E10/mouse) that were 20–50-fold lower than in [132]. Two months after vector administration, C26:0-LPC was unchanged. Cell transduction in the spinal cord or brain was not reported.

The same article reported the results obtained in three groups of *Abcd1^-/y^* mice (*n* = 5–6) injected IT at 9–12 months with larger doses (5.5E10–5.5.E11 vg per mouse) of the same AAV9-CBA-*hABCD1* vector [145]. Among the 22 mice injected with the larger dose, 18 died three to six months after the injection due to very high vector genome content and inflammation in the heart. Cell transduction was not reported in these high-dose mice. S tests were not performed in the mice injected with 5.5.E10–1.6.E11 vg per mouse and euthanized between 11 and 18 months of age. Six months post-vector, C26:0-LPC had decreased by 8–14% in spinal cord samples. The 8% decrease was significant (*p* < 0.05) in response to 1.6.E11 vg per mouse, indicating that the minimal efficacy dose (med_vlcfa_) defined by the lowering effect on VLCFA was 1.6.E11 vg per mouse, only 1/3 of the very toxic 5.5.E11/mouse dose. Instead, the authors used another vgc measurement to calculate another version of the med (“med_vgc_”), based on the minimal dose leading to a significant persistence of vgc in the spinal cord, e.g., 2.E10 vg per mouse [145]. This ten-fold difference between med_vlcfa_ and med_vgc_ should be kept in mind for further comparison with other routes of administration or other vectors. In summary, the AAV9 vector equipped with the CBA promoter showed mitigated results. IV injections showed no positive results that could qualify this route of administration for translation to patients. ICV and IT routes did not transduce the motor cortex from which the cerebrospinal axons originate. The authors honestly pointed out this major limitation of their strategy: “Because the axonal degeneration occurring in *Abcd1^-/y^* mice appears to be of a dying-back pathology, reaching the body of the cortical motor neuron may be necessary to prevent or mitigate disease progression” [145]. On the other side, the abundant transduction of DRG neurons induced by IT bolus could appear encouraging, because spino-cerebellar axons originate from these neurons [132]. However, as stressed before, neurons have very few peroxisomes and no intrinsic VLCFA. This is why we do not expect the expression of ALDP in transduced DRG neurons to rescue the dysfunction of spinocerebellar axons. After ICV injection, 12% of astrocytes and 3% of microglia but no OL and endothelial cells were transduced in the spinal cord. The authors’ conclusions nevertheless sounded optimistic [49,132,134,135].

The orphan disease status of AMN and the hopes placed in gene therapy led the SBT101 vector [AAV9-CBA-hABCD1] to receive fast-track designation from the FDA in February 2022 and orphan drug designation in March 2022, based on results in the *Abcd1^-/y^* mouse model (https://www.businesswire.com/news/home/20220216005313/en/FDA-Grants-Fast-Track-Designation-to-SBT101-the-First-Investigational-AAV-Based-Gene-Therapy-for-Patients-With AdrenomyeloneuropathyAMN, accessed on 1 July 2025) shortly followed by a Phase 1 trial in 16 patients (NCT05394064) [172,173]. The results of this trial, based on a single IT bolus of the SBT101 vector and expected in 2028, will be important to analyze [172].

### 8.3. Gene Therapy Targeting OL

Our group approached the gene therapy of *Abcd1^-/y^* mice with a different objective, e.g., to prioritize the targeting of OL. We tested two strategies, one aiming at preventing the disease by administering the vector soon after birth, the other at trying to rescue incipient motor deficits in aged mice [128,129]. The AAV9 vector used for both experiments comprises the *hABCD1* cDNA driven by the MAG (myelin-associated glycoprotein) gene promoter, known to express genes in OL [174]. We chose the shortest 300 bp form of the native MAG promoter to build the smallest possible cassette [128,129,174].

#### 8.3.1. For Prevention

The AAV9 vector was administered IV to *Abcd1^-/y^* mice at P10, an age when the AAV9 vector can cross the BBB [55]. 

Peroxisomal expression of ALDP was observed in half of OL and one-third of astrocytes in the dorsal, lateral, and ventral white matter of the spinal cord (Figure 2) [129]. Transduction of OL was expected because the myelin-related MAG gene is activated in OL [174]. Transduction of astrocytes was a surprise because these cells were not supposed to activate the MAG promoter [174]. Three weeks post-vector, OL transduction averaged 39% in the dorsal funiculi of the spinal cord (average of cervical, thoracic, and lumbar levels). The first weeks of age are a period of active OL proliferation in the mouse [175,176]. Thus, we postulate that the active cell divisions of OL have diluted the *hABCD1* copies from P10 to W3 and continued to do so until 2 months of age. On the other side, it has been estimated that 50% of OL present in the corticospinal tracts at 2 months of age survive until at least 20 months of age [177]. At 24 months, we found that 9% of spinal cord OL were still expressing *hABCD1*. We speculate that this decrease from 39% to 9% reflects the loss of ~75% of OL between vector delivery at P10 and euthanasia at 24 months of age. Expression was also present in 40% of cerebellar OL at 3 weeks of age, but almost absent in 24-month-old mice treated at P10 [129]. At 3 weeks of age, 13% of spinal cord astrocytes expressed the transgene, ranging from 32% at the cervical level to 5% at the lumbar level [129]. Two years after vector injection, *hABCD1* expression persisted in 12% of spinal cord astrocytes indicating a very slow turnover of this cell population. In rodents, developmental proliferation of astrocytes occurs mostly around birth, followed by differentiation and maturation during the first few weeks [178,179,180]. Expression was absent in spinal cord neurons and microglia and absent or sparse in brain cells, including the motor cortex, at 3 months and 24 months of age [129].

In contrast with the untreated mice, the vector-treated mice did not develop neurological deficits, even at 24 months of age. Indeed, the vector-treated mice (*n* = 10) showed significantly better motricity than untreated *Abcd1^-/y^* mice (*n* = 9) at 24 months of age and showed no significant difference with the age-matched wild-type animals. The C26:0-LPC had decreased by 41% in the spinal cord of the vector-treated mice [129]. The successful prevention of the mouse disease paved the way for human application, now that neonatal diagnosis can identify future AMN patients.

In summary, the successful transduction of spinal cord OL by the AAV9-MAG vector in the neonatal period appeared sufficient to protect spinal cord axons from dysfunction or degeneration until 2 years of age. The contributory role of *hABCD1* transduction of astrocytes remains to be proven.

#### 8.3.2. Another Series of Experiments Attempted to Treat the Disease at the Time of Symptoms in Adult *Abcd1^-/y^* Mice

The same AAV9-MAG vector was injected intra-cisterna magna (ICM) at 18 months of age when *Abcd1^-/y^* mice start losing balance and motricity [128]. One to three months later, vector-treated mice recovered near-normal motor performances, whereas the neurological state continued to deteriorate in untreated mice (Figure 3). Based on this surprisingly rapid effect of the vector, we postulate that the axonopathy of *Abcd1^-/y^* mice was mostly, if not entirely, dysfunctional at this age [128]. More specifically, we postulate that the restored *hABCD1* expression in OL peroxisomes reduces the production of toxic lipids and enhances the supply of energetic substrates to axons. At 24 months, 6 months post-vector administration, *hABCD1* expression persisted in 22% of OL and 22% of astrocytes in the white matter of the dorsal funiculi of the cervical spinal cord. Punctate images of ALDP showed its incorporation into peroxisomes [128] (Figure 2). The unexpected co-transduction of astrocytes (Figure 2) may have played an additional favorable role. No other cell population was found to express *hABCD1*, notably neurons in the motor cortex or the DRG, or microglia [128].

### 8.4. Gene Therapy Targeting Schwann Cells

To rescue peripheral neuropathy, *hABCD1* should be expressed in Schwann cells. In mice and primates, we found that transduction of Schwann cells is possible with the AAV9-MAG vector [181]. A major challenge for human application would be to find a convenient route of administration able to deliver the vector, notably in sciatic nerves, across the blood-nerve barrier [182]. Engineering novel AAV capsids may help maximize therapeutic transgene delivery [183].

### 8.5. Adrenal Targeting Gene Therapy

IV-administered AAV vectors have the potential to bring and express therapeutic transgenes (under ubiquitous promoters) into the adrenal cortex of mice [184] and non-human primates [185]. In AMN patients, the adrenal cortex shows VLCFA accumulation, leading to adrenal insufficiency [186,187]. In *Abcd1^-/y^* mice, the IV injection of the AAV9-CBA-*hABCD1* vector transduced the adrenals. This would pave the way for a correction of adrenal insufficiency in AMN patients.

## 9. Effects of AAV9-*ABCD1* Vectors in Non-Human Primates (NHP)

### 9.1. Efficacy in Spinal Cord

Very few results have been published on the biodistribution of the AAV9-MAG-*ABCD1* vector in the CNS of *Macaca fascicularis*. We tested the ability of this vector to transfect OL in a single primate through an IT injection (2E13 vg/kg). Three weeks after vector injection, a high level of *ABCD1* expression was observed in the white matter of the spinal cord with 58%, 40%, and 30% of OL expressing *ABCD1* at the cervical, thoracic, and lumbar levels. In the cerebellum, 56% of OL expressed vector *ABCD1*. In the white matter of the spinal cord, *ABCD1* expression was found in 33–41% of the astrocytes, 27% in the cerebellum, and less than 2% in the corpus callosum. *ABCD1* expression was not seen in microglia in all the studied regions of the primate CNS. Very few neurons were found to express *ABCD1*.

In a careful study in *Macaca fascicularis*, Vasireddy et al. reported the effect of various doses (1.06 to 3.38 E13 vg per animal) of an AAV9-CBA-GFP vector delivered through various lumbar intrathecal techniques of delivery [145]. Six months post-vector, 25–100% of spinal cord neurons and DRG neurons were found to express GFP. The highest percentages of neuron transduction were reached with the 24 h infusion of the 3.38 E13 dose. The authors reasoned that these levels of transduction «were within the predicted therapeutic range of approximately 23% of neurons, based on previous findings showing that, owing to random X-inactivation, 23–86% of fibroblasts and white blood cells carrying the pathogenic ABCD1 variant lacked ABCD1 expression in asymptomatic women». We believe that this interpretation would only be valid if the population of neurons studied were the sole cause of the pathology. As we believe instead that the pathology has one of its major sources in the peroxisomes of OL, we do not agree with the proposal put forward by Vasireddy et al. It is unfortunate in this respect that OL and astrocyte transduction were not reported in their study. Relevant data, however, were already available in a study by [188] in which AAV9-CBA-GFP vectors (3–4.5E12 vg/animal) were injected intrathecally into *Macaca fascicularis*. Three weeks later, spinal cord immunohistochemistry found GFP expression in 39.2–69.6% of the motor neurons and 16–56.2% of the interneurons, and a high percentage of DRG neurons. Expression was less frequent in astrocytes and rare in oligodendrocytes and microglia. Transduced neurons were also found in the cerebellum and brainstem.

### 9.2. Vector Transduction in Adrenals

In studies conducted for gene therapy of adrenal diseases, the IV injection of AAV9 or AAV5 vectors (E13 vg/kg) carrying a CAG promoter to *Macaca fascicularis* enabled efficient transduction of the adrenal cortex [185]. Also after IT injection, the adrenal cortex was well transduced by AAV9-CBA-*hABCD1* [49]. Following IT injection of the AAV9-MAG vector, adrenal glands did not show any expression in the experiment cited above, as expected from the specificity of expression driven by the MAG promoter.

### 9.3. Safety

Following a single 6 h intrathecal lumbar infusion of *AAV9-CBA-hABCD1* at doses of 7.8E12–7.6E13 vg/animal, *Macaca fascicularis* showed a transient, dose-dependent, mild to moderate increase in alanine aminotransferase, which returned spontaneously to normal by the time of euthanasia [145]. These reassuring results are consistent for AAV9 vectors with those reported in the review by Hudry et al. [189].

In the single primate injected with the *AAV9-MAG-hABCD1* vector cited above, no expression was found in the liver or heart, as expected from the restricted tropism of the MAG promoter. No increase in alanine aminotransferase was observed.

More general aspects regarding safety, regulatory approval, and cost of AAV vectors can be found in [190,191,192].

## 10. Considerations for Future Gene Therapy in AMN Patients

Unlike for most single-gene diseases, in which the cell-type culprit is well identified, a major question for AMN is to decide which cell type(s) should be the preferred target(s) for gene therapy. Ideally, gene therapy would restore *ABCD1* expression in all neuronal and glial cells. If not, it should focus on cells that have a major and lasting effect on disease mechanisms, but the complexity and intricacy of the pathological mechanisms of AMN among the multiple cell types remain a challenging difficulty. Currently, we know of no vector capable of achieving abundant and simultaneous *ABCD1* expression in relevant neurons of the motor cortex, DRG, and glial cells. In addition, systemic routes of administration face the blood–brain barrier, except during the short period following birth. IT delivery of vectors seems to be a convenient route to reach spinal cord, cerebellum, and DRG cells in adult patients.

The objective of gene therapy is to correct crucial deleterious mechanisms at a time when they are still reversible. In this respect, it is fortunate that AMN has a specific biomarker, e.g., a very high plasma concentration of VLCFA or VLCFA metabolites that establishes the diagnosis at birth in the general male population or in a child whose brother has already developed cALD or AMN. This is not the case for other neurodegenerative diseases. This provides a long disease-free interval during which AMN remains subclinical and accessible to prevention. However, because the same *ABCD1* mutation causes either cALD or AMN, the future clinical syndrome cannot be predicted in a yet asymptomatic child. Follow-up with MRI can detect the early cerebral signs of cALD, but not the myelopathy of AMN.

Before entering the gene therapy field specifically, a brief overview of AMN mechanisms at the molecular and cellular levels will help understand the clinical challenges of all therapeutic interventions.

### 10.1. A Brief View of AMN Natural History

AMN mechanisms have been revisited in a recent review [20]. Myelopathy occurs in most, if not all, patients carrying a mutated *ABCD1* gene, except in those who die at a young age. It appears as the inevitable adult form of X-ALD caused by pathological events taking place in the white matter of the spinal cord. There is yet a major gap in knowledge regarding the early pathogenic mechanisms of AMN. An undisputed primary mechanism driving myelopathy is the lack of degradation and accumulation of VLCFA in all cells that normally oxidize VLCFA in their peroxisomes. This is notably the case of OL, by far the main VLCFA producers and degraders. OL could convert undegraded VLCFA into toxic lipids, such as S1P [20], then inoculate these toxic lipids into axons through myelinic channels. Other mechanisms of axonal dysfunction are also possible. First, the VLCFA accumulating in OL could lead to the incorporation synthesis of myelinic VLCFA-rich gangliosides into myelin [73], which could impair signal propagation along long axons. Secondly, the high-energy demand of distal axons may be impacted if OL fails to supply enough fuel to the local oxidative metabolism to be oxidized locally [193]. One can thus suspect that these mechanisms among others could progressively lead to the degeneration and atrophy observed in the long spinal cord tracts at the end stages of AMN. Microglia may contribute, as suggested by their pattern of activation without pro-inflammatory signs in the spinal cord of autopsied AMN patients [135]. This is in sharp contrast with cALD and its pervasive brain inflammation.

### 10.2. Before the First Clinical Manifestations

Some neonatally screened children will develop cALD, usually between 3 and 10 years, but the majority will pass childhood and adolescence without any clinical manifestations. Afterwards, almost all will develop AMN between 25 and 65 years of age. The most definitive intervention would be to restore the expression of the normal *ABCD1* gene in the CNS shortly after birth, e.g., at a time when the immature BBB would allow intravenous AAV vectors to pass. Intervening at an age when body weight is minimal would also have the advantage of a reduced cost due to a low vector dose. Before the first clinical signs of myelopathy are detected, one could also consider administering an AAV vector carrying *hABCD1* in early adulthood. As the BBB is impermeable to vectors beyond the neonatal period, vectors would then have to be administered through an IT route. Several conditions must be met for such gene therapy to achieve long-term efficacy. Our data support that an important condition for success is efficient transduction of inter-fascicular OL in the spinal cord, brainstem, and cerebellum. Another condition, if the vector effect is expected to last for many years, is that the imported *ABCD1* gene copies persist durably in transduced cells.

### 10.3. Gene Therapy for Symptomatic AMN

The rapid rescue observed in 24-month-old *Abcd1^-/y^* mice in response to the AAV9-MAG-*hABCD1* vector (Figure 2) [128] showing that myelopathy may still be reversible at this stage, is encouraging for AMN patients who show incipient symptoms.

We do not forget the 20% of AMN patients who develop cAMN, thus will develop the challenges that the prevention or rescue of brain inflammation and demyelination in a specific separate review.

## 11. Temporary Conclusions

Gene therapy has started to build promising proofs-of-concept in the *Abcd1^-/y^* mouse model (summarized as a Graphical Abstract in Figure 4), but multiple challenges are yet to be solved. Because the success of translation to AMN patients will depend on *ABCD1* transgene expression in pivotal cell types, a deeper knowledge of disease mechanisms and early natural history remains crucial [20]. On the other hand, technical progress will involve the development of cell-tropic capsids and cell-specific promoters able to target well-defined cell populations or subpopulations in the white matter of the spinal cord, brainstem, and cerebellum. The timing of vector injection will have to be aligned with the natural history of AMN.

While prevention may become an attainable goal thanks to the development of neonatal screening, thousands of patients with already installed clinical manifestations of AMN are eagerly waiting for *ABCD1* gene replacement to progress rapidly. Successful intra-CSF routes of administration in *Abcd1^-/y^* mice seem translatable to AMN patients.

A major obstacle to gene therapy for AMN, like for many CNS diseases diagnosed in late childhood or adulthood [194], is that the blood–spinal-cord barrier does not allow IV-injected vectors to pass. Importantly, the efficacy of AAV vector gene therapy in infants affected with SMA shows that high vector doses administered IV early in life can safely cross the human BBB [195,196]. Also, in recent years, the development of viral vectors or nanoparticles capable of crossing the BBB and enhancing CNS gene delivery to selective neuronal or glial cells has shown rapid progress [197,198,199,200,201,202,203,204]. AAV vectors able to target specific cell types in the CNS have emerged [198,205], notably vectors able to transduce the corticospinal tract [206], a major target for AMN gene therapy.

In a more distant future, genomic and epigenome editing may be another approach for AMN therapy. Indeed, X-linked diseases caused by heterozygous mutations of genes on the X chromosome are particularly suitable for epigenome editing. While the reactivation of the mutated allele by epigenome editing is a promising therapeutic approach for AMN, modalities of genetic and epigenetic editing for the CNS are still in their infancy (reviewed in [207]).

## Figures and Tables

**Figure 1 biomedicines-13-01892-f001:**
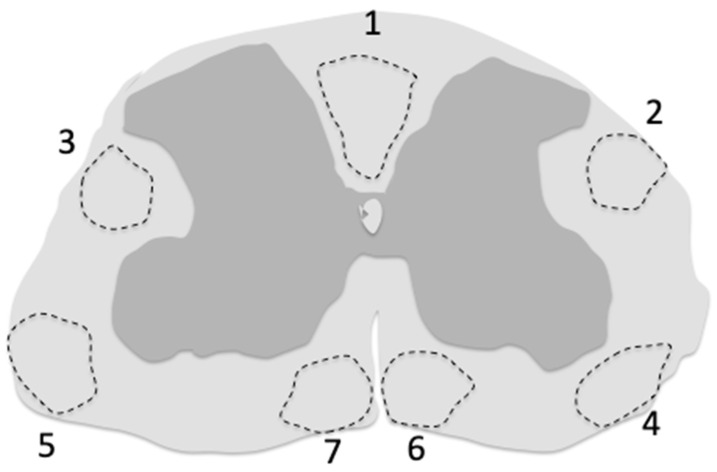
Regions of the cervical spinal cord white matter corresponding to the main ascending and descending tracts affected by AMN pathology. These regions were sampled by Y. Özgür Günes in *Abcd1^y/-^* mice to study cell transduction by the AAV9-MAG-*hABCD1* vector [129]. (1) fasciculus cuneatus, fasciculus cerebrospinalis (pyramidal tract); (2,3) dorsolateral thalamic tract, dorsal spinocerebellar tract, rubrospinal tract; (4,5) ventral spinothalamic tract, ventral spinocerebellar tract; (6,7) ventral spinocerebellar tract, rostral reticulospinal tract.

**Figure 2 biomedicines-13-01892-f002:**
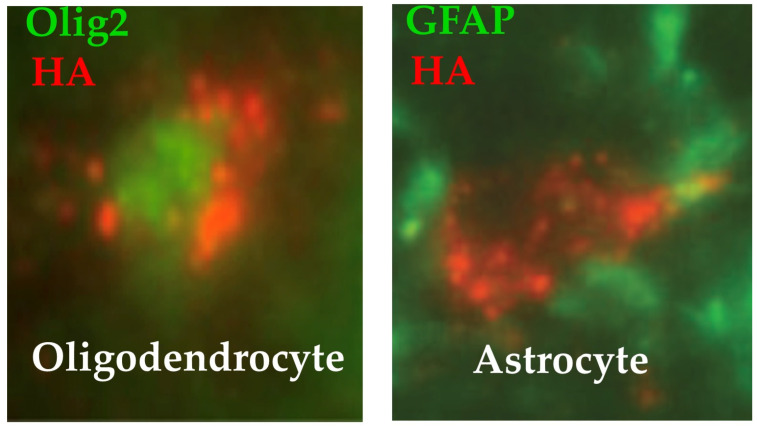
Immunofluorescence images of colocalization of ALDP-HA in peroxisomes and transcription factors in the nucleus of an oligodendrocyte (Olig2) or of an astrocyte (GFAP). Punctate dots indicate that ALDP is localized in peroxisomes [128].

**Figure 3 biomedicines-13-01892-f003:**
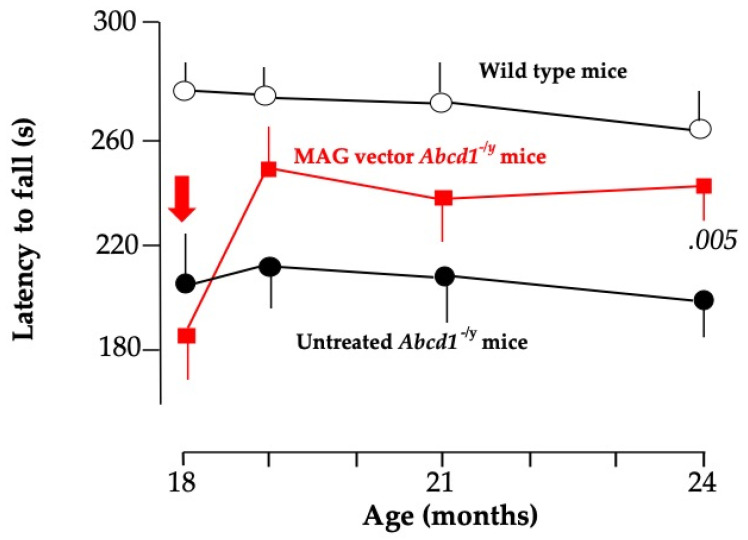
Evolution of the latency to fall from the rotarod in the three groups of mice from 18 to 24 months of age showing the rapid recovery of motor performances in mice treated with the AAV9-MAG vector [135]. *p* < 0.005 vs. untreated *Abcd1^-/y^* mice.

**Figure 4 biomedicines-13-01892-f004:**
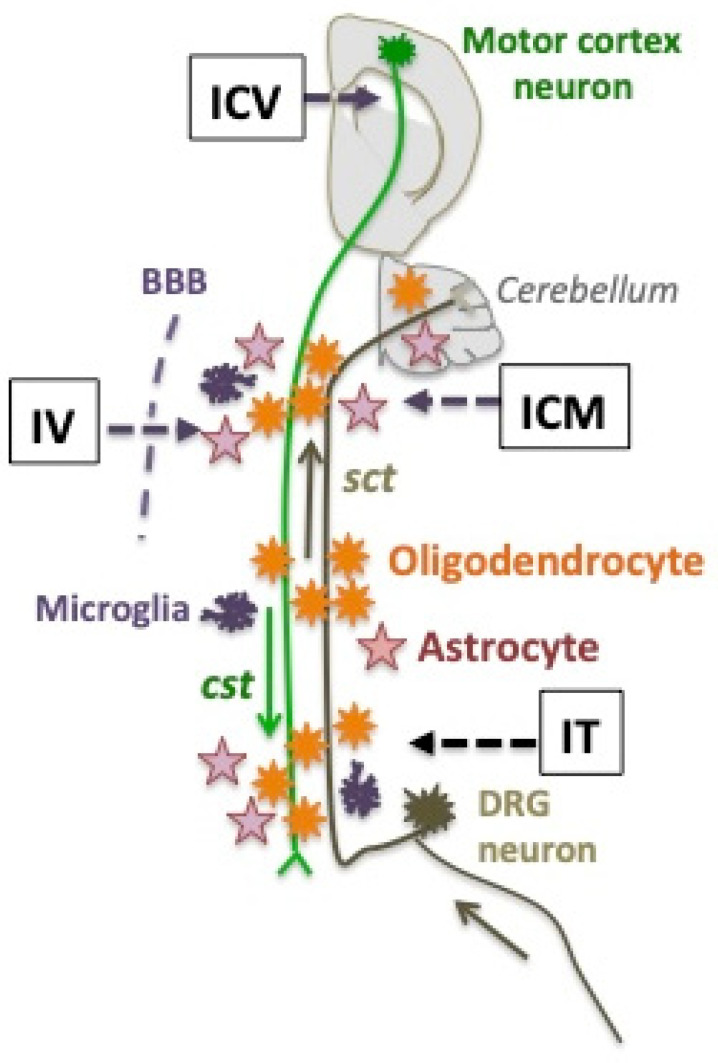
Two long axons traveling the white matter of the spinal cord form the descending corticospinal tracts (cst in green) and the ascending spinocerebellar tracts (sct, in brown). Their respective neurons of origin are in the motor cortex of mouse brain, or in lumbar dorsal root ganglia (DRG). The three main glial cells of the white matter are schematized. The four routes of AAV9 vector administration to *Abcd1^y/-^* mice that have been used are figured as IV (intravenous) in newborns [129] or adults [132,134], IT (intrathecal) in adults [49,132,145], ICM (intra-cisterna magna) in adults [128], or ICV (intra-cerebro-ventricular) in adults [132,134]. Neurons of the motor cortex and microglia were not transduced by any vector. Oligodendrocytes and astrocytes were abundantly transduced by the neonatal IV [129] or the adult ICM [128] administration of the AAV9-MAG vector. DRG neurons and astrocytes were transduced by the IT administration of the AAV9-CAG vector [49].

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
