# Peer review of "Gene Therapy of Adrenomyeloneuropathy: Challenges, Target Cells, and Prospectives"

_biomedicines, 2025, doi:10.3390/biomedicines13081892_

Round 1
Reviewer 1 Report
Comments and Suggestions for Authors
The review goes over the gene therapy approaches aimed at treating AMN using the Abcd1-/y mouse model. It is informative manuscript. However, it requires several major revisions in order to improve clarity, relevance, scientific rigor, and overall contextual quality.
1. References and Literature Update
A high proportion of citations are outdated. Update the reference list with new original studies (2022-2024). Relying on secondary reviews instead of primary research diminishes the rigor of the science. Original experimental data should be prioritized in the references.
2. Structure and Redundancy
Some sections (e.g., cell types involved) are overly detailed or repetitive. Consider summarizing or merging sub-sections for clarity.
The many sub-sections in part 5 (e.g., 5.1–5.7) may disrupt flow; streamlining is recommended.
3. Data Presentation
In final revisions, figures need to be properly described and Include summary tables showing vector type, promoter, route, dose, and outcomes (e.g., transduction % or VLCFA reduction) to improve clarity.
4. Scientific Balance
Although the arguments made regarding the OL-targeted approach are well structured, the manuscript overlooks shortcomings of both OL and neuron-targeted methods.
More thought-out arguments and better critiques of the results and limitations would improve the manuscript.
5. Translational Outlook
Expand on clinical translation challenges, such as vector immunogenicity, delivery barriers, and current status of trials like SBT101.
Address potential safety issues and regulatory matters in more detail.
6. With regard to language
Changing “transfect OL” to “transduce OL” is better in enhancing clarity.
Improving the language encourage better clarity of the text.
Recommendation: Revision: Major
Adjusting the references along restructuring the text, ampler data representation, and adding more thorough discussion would enhance the impact of the influence of the manuscript.
Author Response
The review goes over the gene therapy approaches aimed at treating AMN using the Abcd1-/y mouse model. It is informative manuscript. However, it requires several major revisions in order to improve clarity, relevance, scientific rigor, and overall contextual quality.
1. References and Literature Update. A high proportion of citations are outdated. Update the reference list with new original studies (2022-2024). Relying on secondary reviews instead of primary research diminishes the rigor of the science. Original experimental data should be prioritized in the references.
We did our best to follow Reviewer’s comments. However, unfortunately, few studies in the 2022-2024 have delt with AMN : we have carefully cited them in our review and we believe we have not omitted any recent relevant study. In contrast, many landmark papers were published in the early decades of AMN research.
In our previous version, 26% of the cited references were published on AMN between 2022 and 2024. This proportion is comparable with many published reviews. For example, the percentage of publications dated during the 2-3 years preceeding the writing of the review was calculated.
- Keil, et al. (2025) Science, 389 (6756), eadv8269. (34%)
- Noor et al. (2025). Biomedicines, 13(6), 1426 (27%)
- Simons, M., et al. (2024). Cold Spring Harbor perspectives in biology, 16(10), a041359 (15%).
As asked by the Reviewer, we have prioritized original articles but also had to cite landmark reviews written by AMN experts.
The policy of Biomedicines being to exclude publications with corrigenda, it makes impossible to cite several of excellent 2022-2024 original publications in Nature, Nature Com, etc. This may not be justified when corrigenda are about rather small changes, sometimes entirely formal, but we had no other choice than respecting editorial policy.
2. Structure and Redundancy. Some sections (e.g., cell types involved) are overly detailed or repetitive. Consider summarizing or merging sub-sections for clarity. The many sub-sections in part 5 (e.g., 5.1–5.7) may disrupt flow; streamlining is recommended.
We agree with the Reviewer that it was entirely justified to re-write some sections, notably those on cell types, to clarify and correct a relatively high degree of redundancy in the previous version. We have however kept the frame of 5.1-5.6 subsections, because we think they will help readers to refer to each distinctive cell type category.
3. Data Presentation: in final revisions, figures need to be properly described and include summary tables showing vector type, promoter, route, dose, and outcomes (e.g., transduction % or VLCFA reduction) to improve clarity.
We did try to do that in the first version. However, the studies were highly heterogeneous for vector nature and dose, routes of vector administration, ages at intervention, duration of follow-up before euthanasia, behavioral phenotypes, mode of calculation of transduced cells. Thus we could not make a clear summary table recapitulating the different gene therapy studies. Because all details matter, we had to dissect out the studies to allow a precise description and comparison across studies.
4. Scientific Balance
Although the arguments made regarding the OL-targeted approach are well structured, the manuscript overlooks shortcomings of both OL and neuron-targeted methods. More thought-out arguments and better critiques of the results and limitations would improve the manuscript.
We fully agree thus have followed Reviewer’s suggestion in the new version with regard to critiques and limitations of the two main approaches, expressed more clearly and more strongly.
Translational Outlook. Expand on clinical translation challenges, such as vector immunogenicity, delivery barriers, and current status of trials like SBT101.
We did it in the revised version. We cited several master reviews on these topics of AAV gene therapy, because it was not possible to go into a detailed analysis of them in a review specifically devoted to AMN.
The results of the SBT101 trial are yet embargoed, thus we had no scientific information to communicate. We are not aware of other trials in AMN.
- With regard to language. Changing “transfect OL” to “transduce OL” is better in enhancing clarity.
Improving the language encourage better clarity of the text.
We agree and have done our best for the clarity of text in the revised version, working with an american scientific writer in biology.
Recommendation.
Adjusting the references along restructuring the text, ampler data representation, and adding more thorough discussion would enhance the impact of the influence of the manuscript.
We have modified more than 20% of the text according to Reviewer’s recommendation and Editorial Policy. We thank him for pushing us to restructure redundant parts of the text, express our views more critically. Changes are too many to be reported. See for example the “Non-human-primate Section”.
Reviewer 2 Report
Comments and Suggestions for Authors
In the current review paper, the authors summarized the detailed the pathology of X-ALD and AMN and detailed the progresses in gene therapy research delivering the ABCD1 transgene to various target tissue types of interest. While the paper has been well-written and contains a very comprehensive summary for the topic, it may benefit from additional edits to promote readership. A few suggestions:
- The title of the paper appears to be a bit narrower than the scope of the entirety of the review. A new title that encompasses challenges, past and on-going research directions and future prospectives could potentially better help the readers navigate the contents to expect in the main texts.
- The Abstract appears a bit too lengthy. Particularly, in light that this is a review paper, the sentences led by “Methods”, “Results” and “Conclusion” do not seem to be fitting to the theme of the review article. Those contents could be compacted to more concise language that invite the readers to the detailed discussions placed later in the main text.
- It could be helpful to discuss a bit more about the population genetics data on the ABCD1 registry to guide the readers on the nature and distribution of the pathological mutations across the gene and its exons.
- Line 137: “adeno-associated viral (AAV)” should be changed to “AAV” as its full name has been spelt out early-on in the article.
- Line 154: “a” should be changed to “an”.
- Sub-sections titled “Adrenal Insufficiency” and “a brief view of AMN Natural History” do not seem to belong under the Section “Cell types involved in AMN Pathology”.
- Line 345: the title might be more appropriate to have “How Evaluate” changed to something like “Evaluation criteria for”.
Author Response
In the current review paper, the authors summarized the detailed the pathology of X-ALD and AMN and detailed the progresses in gene therapy research delivering the ABCD1 transgene to various target tissue types of interest. While the paper has been well-written and contains a very comprehensive summary for the topic, it may benefit from additional edits to promote readership.
We thank the Reviewer for his positive comments, and for the following suggestions notably (3).
A few suggestions:
- Line 137: “adeno-associated viral (AAV)” should be changed to “AAV” as its full name has been spelt out early-on in the article. Done
- Line 154: “a” should be changed to “an”. Done
- Sub-sections titled “Adrenal Insufficiency” and “a brief view of AMN Natural History” do not seem to belong under the Section “Cell types involved in AMN Pathology”. This was corrected and moved to 10.1, respectively.
- Line 345: the title might be more appropriate to have “How Evaluate” changed to something like “Evaluation criteria for”. Corrected.
Reviewer 3 Report
Comments and Suggestions for Authors
The article 'Challenges in the Gene Therapy of Adrenomyeloneuropathy' is an in-depth, high-quality review of modern AMN gene therapy approaches that is also up to date. The authors present a detailed analysis of the disease’s pathogenesis, the involvement of various cell types, and experimental strategies based on the Abcd1-/y model. They also explore the potential for clinical translation. Of particular value is the emphasis placed on oligodendrocytes as key therapeutic targets, a point supported by the authors' own preclinical data.
The work is distinguished by its logical structure, critical approach to previously published data and clear argumentation. This review will be useful for researchers and developers of therapies for rare neurodegenerative diseases. Despite the technical detail being somewhat overwhelming at times, this article is a valuable resource that will contribute to the further development of effective gene therapies for adrenomyeloneuropathy.
Comments for the authors:
- It has been shown that the level of ABCD1 expression in oligodendrocytes transduced at neonatal age decreases significantly by 24 months. Are there any plans to increase the stability of expression, for example, through the use of integrating vectors or repeated administrations?
- Did the authors consider combination therapy, such as gene therapy targeting oligodendrocytes alongside immunomodulation, to enhance the neuroprotective response of microglia?
- It would be useful to include an additional schematic illustration summarizing the comparative efficacy of different AAV vectors, promoters, and routes of administration (IV, IT, ICM, etc.), as well as their targeting of specific cell types. Such a diagram (e.g., in the form of a table or infographic) would make the article more visual and useful for readers planning their own gene therapies for adrenomyeloneuropathy or similar diseases.
Author Response
The article 'Challenges in the Gene Therapy of Adrenomyeloneuropathy' is an in-depth, high-quality review of modern AMN gene therapy approaches that is also up to date.
We thank the Reviewer for appreciating our effort to update our review and deliver high quality information
The authors present a detailed analysis of the disease’s pathogenesis, the involvement of various cell types, and experimental strategies based on the Abcd1-/y model.
Discussing target cell types and gene therapy strategies was our main objective.
They also explore the potential for clinical translation.
As already pointed out by another Reviewer, the potential for clinical translation inspired our analysis of mouse studies.
Of particular value is the emphasis placed on oligodendrocytes as key therapeutic targets, a point supported by the authors' own preclinical data.
Again, we appreciate the comment of the Reviewer about our recognition of oligodendrocyte role and the review of our own studies, which could be discussed here in a more general context than in the specific original research article.
The work is distinguished by its logical structure, critical approach to previously published data and clear argumentation.
We are very happy that our efforts were recognized.
This review will be useful for researchers and developers of therapies for rare neurodegenerative diseases.
This was one of our major objectives.
Despite the technical detail being somewhat overwhelming at times, this article is a valuable resource that will contribute to the further development of effective gene therapies for adrenomyeloneuropathy.
The review offered an opportunity to present and discuss technical details. While we agree that details were “overwhelming at times”, these details could make big differences across studies. For example, counting cell transduction only 3 weeks after vector instead of 6 months, does not allow to reliably figure which percentage of cells keeps transgene copies in the mid or long term. Also, the review was an occasion to critically recapitulate the relevant criteria of successful gene therapy regarding vector doses, timing vs natural history of disease, routes of delivery, promoter differences, or transduced cells. We could not do this without presenting details of procedures and results.
Comments for the authors:
1. It has been shown that the level of ABCD1 expression in oligodendrocytes transduced at neonatal age decreases significantly by 24 months. Are there any plans to increase the stability of expression, for example, through the use of integrating vectors or repeated administrations?
We have discussed this point precisely in the revised version, showing that the stability of expression is mostly conditioned by the proliferation of the various targeted cell types. Given the lifespan of mouse (30 mo), and the different lifespan of transduced cells (neurons, astrocytes, oligodendrocytes) we considered that reaching 24 months of age with near-normal motricity indicates that neonatal injections of vector could cover the natural history and pathological mechanisms of mouse AMN, even if percent of cells transduced went from 39 to 9% . AMN axonal degeneration is a very progressive process; if it is interrupted soon enough, our belief is it will not have enough time to recur. In adult mouse, the vector injected at 18 months was still present in many oligodendrocytes and astrocytes 6 months later.
Re-dosing is a potential procedure of gene therapy, discussed in several reviews. We do not think integrative vectors are an attractive option given the potentially serious side effects of uncontrolled integration site on the genome.
2. Did the authors consider combination therapy, such as gene therapy targeting oligodendrocytes alongside immunomodulation, to enhance the neuroprotective response of microglia?
We agree with the Reviewer, but we did not have any solid experimental data to substantiate i) the role of microglia in AMN ii) the capacity to transduce these resident yolk-sac derived cells iii) the effect of immunomodulation on microglia
3. It would be useful to include an additional schematic illustration summarizing the comparative efficacy of different AAV vectors, promoters, and routes of administration (IV, IT, ICM, etc.), as well as their targeting of specific cell types. Such a diagram (e.g., in the form of a table or infographic) would make the article more visual and useful for readers planning their own gene therapies for adrenomyeloneuropathy or similar diseases.
We fully agree and we attempted to make the article more visual as suggested by the Reviewer, as an illustration presented at Figure 4. We were unable to make a clear summary table because of the heterogeneity of experimental conditions and findings.
Round 2
Reviewer 1 Report
Comments and Suggestions for Authors
Revisions satisfactory.
Accepted